# Collective dynamics support group drumming, reduce variability, and stabilize tempo drift

Dobromir Dotov[1,2,3]*, Lana Delasanta[4,5], Daniel J Cameron[1,2], Edward W Large[4,5,6,7], Laurel Trainor[1,2,8]

[1]LIVELab, McMaster University, Hamilton, Canada; [2]Psychology, Neuroscience and Behaviour, McMaster University, Hamilton, Canada; [3]Department of Biomechanics, University of Nebraska Omaha, Omaha, United States; [4]Center for the Ecological Study of Perception and Action, University of Connecticut, Storrs, United States; [5]Department of Psychological Sciences, University of Connecticut, Storrs, United States; [6]Department of Physics, University of Connecticut, Storrs, United States; [7]CT Institute for Brain and Cognitive Sciences, University of Connecticut, Storrs, United States; [8]Rotman Research Institute, Toronto, Canada

**Abstract** Humans are social animals who engage in a variety of collective activities requiring coordinated action. Among these, music is a defining and ancient aspect of human sociality. Human social interaction has largely been addressed in dyadic paradigms, and it is yet to be determined whether the ensuing conclusions generalize to larger groups. Studied more extensively in non-human animal behavior, the presence of multiple agents engaged in the same task space creates different constraints and possibilities than in simpler dyadic interactions. We addressed whether collective dynamics play a role in human circle drumming. The task was to synchronize in a group with an initial reference pattern and then maintain synchronization after it was muted. We varied the number of drummers from solo to dyad, quartet, and octet. The observed lower variability, lack of speeding up, smoother individual dynamics, and leader-less inter-personal coordination indicated that stability increased as group size increased, a sort of temporal wisdom of crowds. We propose a hybrid continuous-discrete Kuramoto model for emergent group synchronization with a pulse-based coupling that exhibits a mean field positive feedback loop. This research suggests that collective phenomena are among the factors that play a role in social cognition.

*For correspondence:
ddotov@unomaha.edu

Competing interest: The authors declare that no competing interests exist.

## Editor's evaluation

Taking joint drumming as a model of collective dynamics, and combining solid quantitative methods, the authors characterize how human behavior changes, at the individual- and group-level, as a function of group numerosity. A take-home message of this important work is that not everything we know from studies involving dyads should be necessarily generalized to larger groups. This study will be of great interest to scientists looking for new approaches to understanding group behavior, especially within the fields of human cognition, neurosciences, and musicology.

## Introduction

Humans are social animals who engage in a variety of collective activities requiring coordinated joint action. Collective goals can be achieved through spontaneously distributed workloads among group

members, such that the emerging collective dynamics confer benefits to performance not available to individuals—the 'wisdom of crowds' (*Galton, 1907*; *Surowiecki, 2005*).

Principles of collective dynamics explain the adaptive value of collective behavior in some species (*Couzin, 2018*). For example, collective dynamics can overcome the limitations of individuals' knowledge and ability to communicate (*Goldstone and Roberts, 2006*) by integrating information quickly during group decision making (*Ispolatov, 2015*; *Miller et al., 2013*; *Rosenthal et al., 2015*). Large coherent swarming and flocking in groups can arise from short-range interactions among proximal individuals in non-human animals including primates (*Farine et al., 2017*), insects (*Jacobs et al., 2007*), fish, and birds (*Miller et al., 2013*; *Parrish et al., 2002*). As such dynamics have been demonstrated in numerous species, theoretical models have been developed to provide additional quantitative explanatory support. In contrast, in spatial tasks such as navigation, aggregate group behavior, formalized as a mean field, can serve to stabilize feedback to individuals by virtue of being a group average (*Berdahl et al., 2018*; *Sumpter, 2006*; *Torney et al., 2009*). Here, we test whether mean field behavior accounts for stability of temporal group dynamics in humans during a drumming task.

Collective dynamics are observable in animal as well as in human group behavior. Crowds of walking individuals achieve globally coherent states based on local inter-individual interactions (*Rio et al., 2018*; *Warren, 2018*). In audiences, individual-group interactions and social contagion govern the spontaneous onset and offset of applause (*Mann et al., 2013*). Collective dynamics can also play a constitutive role in sports by permitting advantages not available to individuals alone (*Vilar et al., 2013*). Evidence of collective dynamics is seen even in the minimal group, a dyad: spontaneous synchronization between two individuals emerges from constraints such as weak coupling (*Oullier et al., 2008*; *Schmidt et al., 1990*; *Schmidt and O'Brien, 1997*).

Synchronized group action is an essential element of music making, a defining social behavior of human interaction (*Honing et al., 2015*; *Patel and Iversen, 2014*; *Salimpoor et al., 2011*; *Savage et al., 2015*; *Trainor, 2015*), and the value of precise synchronization may vary from culture to culture and across musical contexts (*Benadon et al., 2018*; *Davies et al., 2013*; *Lucas et al., 2011*).The archaeological evidence of musical instruments goes back 30,000 years, and singing and drumming are thought to be even older (*Conard et al., 2009*). The evolutionary origins of musical rhythmic actions may relate to social motor behaviors in non-human species, such as the synchronization and desynchronization of vocalizations between individuals in group chorusing, arising from pressures to either collaborate or compete (*Gamba et al., 2016*; *Greenfield et al., 2017*; *Ravignani et al., 2014*; *Ravignani et al., 2019*). Fundamental aspects of music are present in humans from early stages in development: infants show early musical preferences, social-emotional responses to music, and rate-sensitive motoric responses to musical rhythm (*Cirelli et al., 2014*; *Cirelli et al., 2018*; *Trainor and Marsh-Rollo, 2019*; *Zentner and Eerola, 2010*). Although musical behavior is found in individuals alone, it occurs mainly in groups ranging from duets to hundreds of participants. Yet, the role of collective dynamics, especially in groups larger than dyads, has largely remained untested. Here, we consider a musical task in which timing consistency and synchrony are crucial. We investigate in what ways the group average performs better than the individuals, a temporal version of the wisdom of the crowd phenomenon.

To account for group interaction, one popular strategy is to first develop a theory for single-person tasks and then extrapolate into dyadic contexts. For example, the idea that the brain is a prediction machine that enables timely motor control in an uncertain environment can be extrapolated as two brains mutually predicting each other (*Friston and Frith, 2015*; *Wolpert et al., 2003*). This means that each individual instantiates two processes: the first for controlling one's own rhythm and timing and the second for predicting the rhythm and timing of the partner in the dyad (*Heggli et al., 2019*; *Heggli et al., 2021*; *Keller et al., 2014*; *van der Steen and Keller, 2013*). It is not clear, however, if this can serve as an adequate foundation for behavior in larger groups. Every individual would have to predict every other individual and potentially factor-in higher-order predictions. Furthermore, at the perceptual level, group size changes the auditory information available to an individual. A listener is likely to hear an entire choir as one, or a small number of, coherent sounds rather than perceive every singer's voice within that choir.

Group interaction can also be addressed in terms of theoretical models explicitly developed for synchronization in systems made of many dynamic units (*Alderisio et al., 2016*; *Oullier et al., 2008*; *Schmidt et al., 1990*; *Schmidt and O'Brien, 1997*). The individual units in such models usually do

not contain processes dedicated separately to self-timing and other-predicting. Instead, each unit has only a self-timing and a phase-correction coupling term. The propagation of phase adjustment across all units is sufficient for the collective to enter a group-synchronized state, despite that each unit is only trying to cancel a phase difference. An important benefit of this approach is that it is inherently collective, converging on the same formalisms used for small or large groups of animals (*Kelso, 2021*; *Zhang et al., 2019*). Here, we consider such a system of coupled oscillators to account for empirical data on individual and group variability in a drumming task.

Group music making constitutes an ecologically valid and convenient paradigm for studying group action and collective experience in the laboratory (e.g. *D'Ausilio et al., 2015*). We used a timing task performed by groups of different sizes. Larger group dynamics are less studied because measuring highly precise timing while collecting group data is difficult for both logistical and methodological reasons (cf. *Alderisio et al., 2016*; *Chang et al., 2017*; *Chang et al., 2019*; *Chauvigné et al., 2019*; *Shahal et al., 2020*). Participants completed a group synchronization-continuation task (SCT; *Figure 1A*). It required them to drum in synchrony with an isochronous auditory stimulus and continue drumming at the same rate after the stimulus stopped, while we collected the onset times of each drum hit (*Figure 1B*). Stimulus tempo was varied across trials. We tested groups of two (dyads), four (quartets), and eight (octets) participants (*Figure 1C*). The task tested participants' ability to synchronize to an external reference as well as to other participants, while also minimizing temporal variability and maintaining the initial stimulus rate. This ensemble drumming was also compared against the solo condition. Specifically, participants tested in duets or quartets completed both solo and ensemble conditions. The octet group did not complete the solo conditions for logistical reasons, but they instead completed a control condition in which the synchronization phase continued for the whole trial (i.e. there was no continuation phase) as well as trials using a more complex and musically realistic rhythm. Analyses of the more complex rhythm will be reported separately.

Inspired by theoretical models from the animal literature (*Sumpter, 2006*), we assumed that individual interactions average to overall group behavior, a mean field, which provides stabilizing feedback to individual group members (*Figure 2*). A key prediction was that the relative influence of the stabilizing feedback would increase with increasing ensemble size. To this end, we measured separately the variability of individual drummers and the group. We constructed a theoretical group-aggregate onset time as the center of clusters of individual onset times. We expected that larger groups would exhibit lower variability as measured using the coefficient of variation of inter-onset intervals. We also tested this idea formally by adapting a Kuramoto model of group synchronization.

The (*Kuramoto, 1975*) dynamic system of coupled phase oscillators, *Equation 1*, was conceptualized as a large population of oscillators with different natural frequencies capable of spontaneously locking to a common frequency.

$$\dot{\theta}_i = \omega_i + \frac{K}{N} \sum_{j=1}^{N} \sin\left(\theta_j - \theta_i\right) \tag{1}$$

Here, $\theta_i$ is the phase of oscillator $i$, $\omega_i$ is its preferred frequency, i.e., how fast around the unit circle it likes to go, $K$ is coupling strength, and $N$ is the number of oscillators.

The model gives a mathematical account of group synchronization as dependent on a mean field, referred to here as group aggregate. A central feature of the model is that the feedback is positive: the amplitude of the mean field grows as a function of inter-individual synchronization, and reciprocally, the individual oscillators are affected more by the mean field if its amplitude is larger, see *Figure 2*. This is shown by using the definition of the mean of phases, *Equation 2*, to express *Equation 1* equivalently (*Strogatz, 2000*) in terms of the coupling between individual oscillators and the mean field, *Equation 3*. $\Psi$ is the mean field phase, and $r$ is the mean field coherence, also called order parameter (*Kuramoto, 1975*).

$$re^{i\Psi} = \frac{1}{N} \sum_{j=1}^{N} e^{i\theta_j} \tag{2}$$

$$\dot{\theta}_i = \omega_i + rK \sin\left(\Psi - \theta_i\right) \tag{3}$$

Such individual-collective positive feedback loops enable ant trails and other phenomena in swarming animals (*Sumpter, 2006*) as well as acoustic herding in chorusing animals (*Ravignani et al.,*

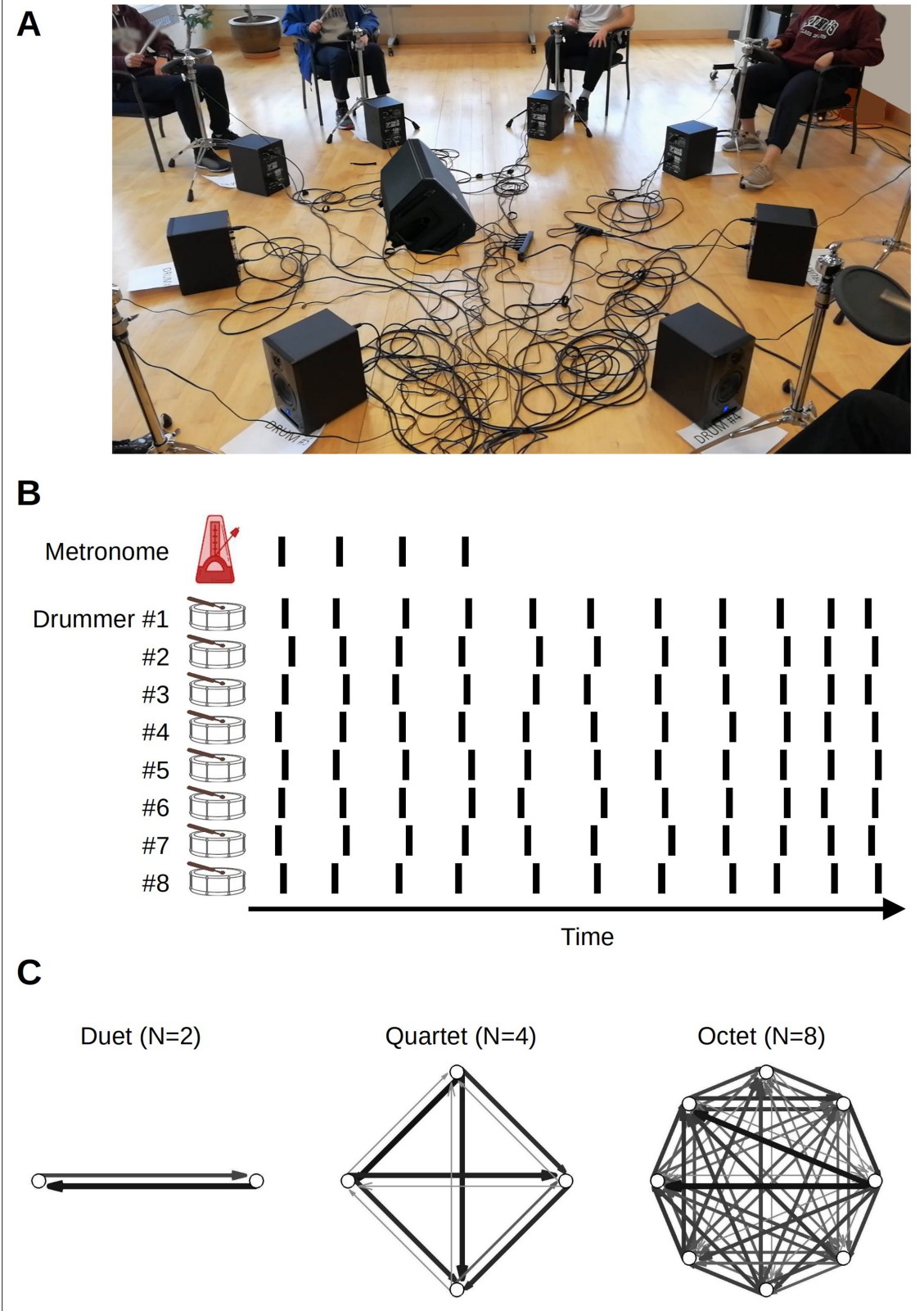

**Figure 1.** Experiment setup. (**A**) In ensemble condition, drummers faced each other in a circle. (**B**) The main task was synchronization-continuation where participants were paced initially by an auditory stimulus and then had to maintain the rhythm, tempo, and synchronization among each other. The inter-onset intervals between drum hits, shown schematically as vertical lines, were used to obtain individual-level measures of variability and speeding up. Cross-correlation and transfer entropy were used as pair-level measures of synchronization and interaction. (**C**) Transfer entropies, color, and width-coded, from three sample trials from different groups. Network analysis was applied to these graphs to obtain group-level characterization.

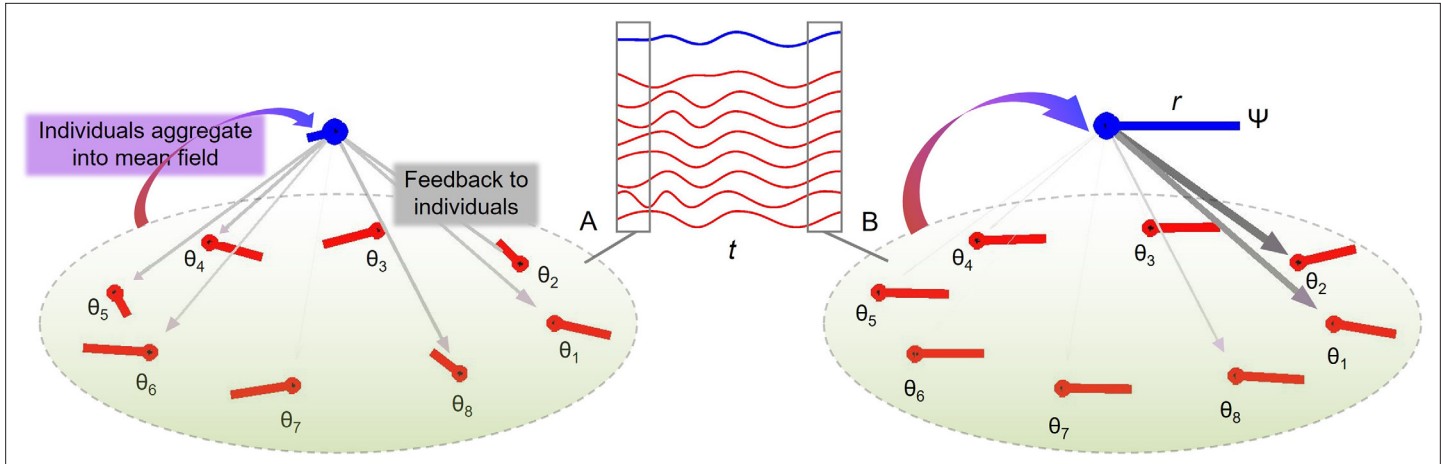

**Figure 2.** Group synchronization of eight oscillators with random initial conditions (**A**) and coherent phases later in the trial (**B**). The middle inset shows individual trajectories (red lines) and the mean field (blue line) in time (t). Averaging the phase oscillators $\theta$ (red lines) gives a so-called mean field (blue lines) with phase $\Psi$ and amplitude $r$. As the individual oscillators become more coherent from (**A**) to (**B**), $r$ increases which leads to stronger influence from $\Psi$ to diverging $\theta$'s (weight of the downward vectors).

The online version of this article includes the following figure supplement(s) for figure 2:

**Figure supplement 1.** The gamma distribution (a=1.25 and b=.02) in the pulse-coupled model of group synchronization localizes the coupling in time, **Equation 5**.

2014). The system also serves as a model for neuronal collective synchronization (**Breakspear et al., 2010**; **Frank et al., 2000**; **Noori et al., 2020**). Theoretically, the principles embodied by this system should apply to group action in humans too (**Zhang et al., 2019**). It has found application in understanding inter-personal synchronization of dyads (**Dotov et al., 2019**; **Heggli et al., 2019**; **Roman et al., 2019**), individual rate preferences in the dyad (**Bégel et al., 2022**), and effects of coupling topology in larger groups (**Alderisio et al., 2016**). In the present context, the model predicts that ensembles will be more stable than individuals because of the feedback loop between the timing of individuals and the group aggregate of individuals.

The original Kuramoto model involves continuous dynamics and continuous coupling with constant gain and no delay. In a drumming task, participants are coupled by way of discrete acoustic onset times. Here, we propose a hybrid continuous-discrete system of oscillators with event-based feedback updated once per cycle, **Equation 4**. It utilizes a pulse function shaped like the acoustic envelope of a drum hit. To this end, **Equation 5** specifies an asymmetric gamma distribution that rises sharply at time 0 and then decays slowly (**Figure 2—figure supplement 1**). In the model, an oscillator emits a pulse once per cycle when its phase angle crosses zero. Furthermore, we posit additive phase variability with a Gaussian distribution $N(0, \sigma)$.

$$\dot{\theta}_i = \omega_i + \frac{K}{N} \sum_{j=1}^{N} P_{j \neq i}(t) \sin\theta_i + \mathcal{N}(0, \sigma), \cdots\cdots i = 1, ..., N \quad (4)$$

$$P(t) = f(t|a, b) = \frac{1}{b^a \Gamma(a)} t^{a-1} e^{\frac{-t}{b}} \quad (5)$$

Like the original system, this model involves intrinsic dynamics ($\omega_i$) and interaction dynamics given by the whole coupling term. For comparison, we tested two additional models: the classic Kuramoto system with constant full coupling, **Equation 1**, as well as the hybrid continuous-discrete model, **Equations 4; 5**, with a sparse coupling matrix in a ring topology. Specifically, the individual units were only coupled to their two immediate neighbors, $j = i \pm 1$, with a periodic boundary condition, $\theta_{N+1} = \theta_1$ and $\theta_0 = \theta_N$.

# Results

In Performance, Dynamics and coordination in ensemble, SCT conditions, and Group dynamics, we present the analyses of the participants' drumming behavior. In Variability in the mean-field model with discretely updated feedback, we present the results from our adapted hybrid model consisting of a continuous-discrete system of oscillators with pulse-based feedback updated once per cycle.

## Performance

### Variability

#### Does individual variability increase in ensemble compared to solo conditions?

To answer this question, we fitted a model including only individuals' data in dyads and quartets (octets did not complete solo conditions). As *Figure 3A* shows, individuals' variability increased when playing in an ensemble relative to playing solo ($\beta$=0.009, SE=0.001, t=6.65, and $\omega^2$=0.10).

#### In the context of true ensemble playing, how does variability of the individuals and group aggregate change across dyads, quartets, and octets?

To specifically test the main hypothesis, we fitted a model to continuation (SCT) trials from all three group sizes in the ensemble playing condition. We found that individual variability increased with group size ($\beta$=0.001, SE=0.0005, t=2.20, and $\omega^2$=0.09), group-aggregate variability was lower than individual variability ($\beta$=−0.010, SE=0.003, t=−3.29, and $\omega^2$=0.01), and the difference between individual and group-aggregate variability increased for larger group sizes ($\beta$=−0.0025, SE=0.0005, t=−5.19, and $\omega^2$=0.03), see *Figure 3A*. As a sanity check, we verified that the procedure for group-aggregate onset times did not lead to spuriously periodic data. We observed very high variability for the pseudo-group-aggregate in solo condition where individuals did not hear each other, see *Figure 3—figure supplement 1A*.

#### How does variability compare between continuation and synchronization?

The octet group completed trials in conditions of both SCT (i.e. after the offset of the reference metronome) and synchronization-only (with a constant reference metronome). Surprisingly, there were no differences between these conditions either in individual (t<1) or in group-aggregate variability (t<1).

### Speeding up

#### Do individuals speed up more when playing solo than in dyad or quartet ensembles?

Speeding up was defined as the linear increase in tempo (i.e. decrease of IOIs) over the course of a trial. As expected, individuals sped up even when playing solo ($\beta$=0.018, SE=0.007, t=2.67, and $\omega^2$=0.10), and greater speeding up was observed in ensemble than in solo conditions ($\beta$=0.022, SE=0.007, t=3.29, and $\omega^2$=0.16). The effect of group size was not significant (t<1).

#### Do dyad, quartet, and octet ensembles speed up to different extents?

To test the effect of group size across the duets, quartets, and octets, a second model was fitted with group size as a continuous predictor, but only including trials in SCT ensemble playing. As suggested by *Figure 4*, there was a trend for speeding up to decrease as group size increased. The best linear model included an effect of tempo ($\beta$=−0.001, SE=0.00005, t=−1.98, and $\omega^2$=0.005) and an interaction between group size and tempo ($\beta$=−.00004, SE=0.000007, t=−6.20, and $\omega^2$=0.11), reflecting that at higher tempos larger ensembles sped up less than smaller ensembles, see *Figure 4—figure supplement 1*.

## Dynamics and coordination in ensemble, SCT conditions

### Individual and group-aggregate dynamics (autocorrelations)

#### Individuals

*Figure 5A* shows that individuals' autocorrelations at lag 1 were negative across all group sizes and tempos. This is typical for the alternating long-short IOIs seen in synchronization tasks, reflecting

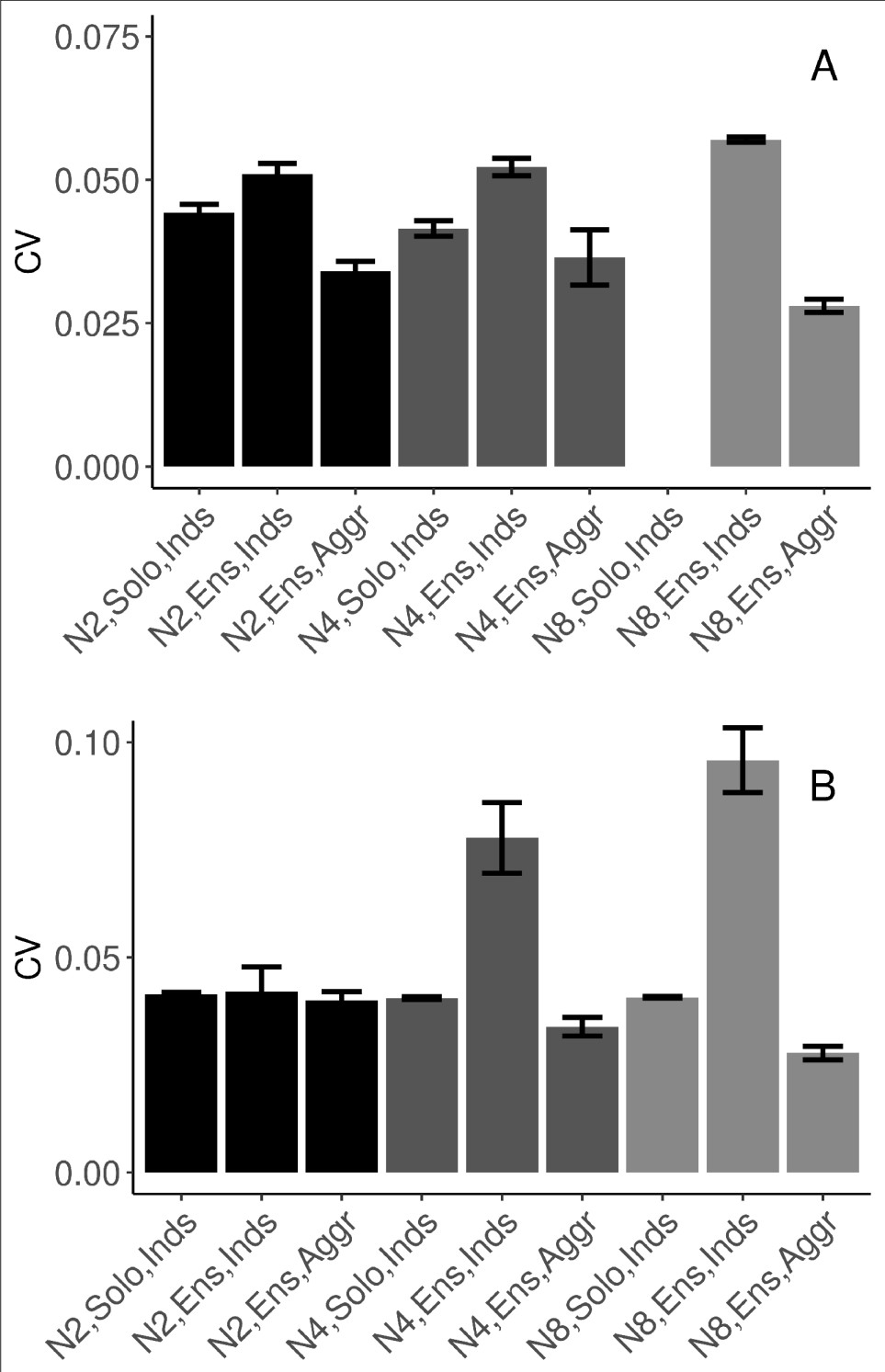

**Figure 3.** Task performance measured in terms of mean (SE) coefficient of variation of inter-onset intervals (IOIs). (**A**) Synchronization-continuation task (SCT) drumming trials (no data collected in N8 solo condition). (**B**) Pulse-coupled Kuramoto model. N2=dyad; N4=quartet; N8=octet; Inds = individual participants; Aggr = group-aggregate; Solo = solo condition; Ensemble = ensemble condition. Error bars are standard errors. With missing trials, the number of observations in the respective conditions was $n$=(88, 88, 44, 111, 110, 32, Ø, 953, 133) in (A), and $n$=(200, 198, 99, 400, 400, 100, 800, 688, 86) in (B).

The online version of this article includes the following source data and figure supplement(s) for figure 3:

*Figure 3 continued on next page*

*Figure 3 continued*

**Source data 1.**

**Figure supplement 1.** Task performance measured in terms of mean (SE) coefficient of variation of IOIs.

**Figure supplement 2.** Variability in alternative models of group synchronization.

alternating fast-slow synchronization errors when adapting to the previous interval. There was also a trend across all group sizes and tempo conditions for positive peaks at even lags (2, 4, 6, and 8) and negative peaks at odd lags (3, 5, and 7), see *Figure 5—figure supplement 1*. However, this general pattern became smoother with increasing group size. To test this statistically, in each trial, we took the average absolute difference between successive lags, thus measuring the average range of the auto-correlation function up to lag 8 and applied the same linear modeling approach as in Performance. The effect of group size was significant ($\beta=-0.0054$, SE=0.0023, $t=-2.35$, and $\omega^2=0.13$). There was also a decrease of range with increasing tempo ($\beta=-0.00039$, SE=0.0001, $t=-4.92$, and $\omega^2=0.04$). Additionally, we verified the signs, relative magnitudes, and the effects of group size and tempo on the autocorrelations by fitting separate linear models for each lag, see *Supplementary file 1a*.

## Group-aggregate

Interestingly, group-aggregate IOIs revealed the same overall pattern of results even though by definition the group-aggregate timing was smoother and less variable (see Group-aggregate (mean field) measure for ensembles and pseudo-ensembles), see *Figure 5B* and *Supplementary file 1b*. There

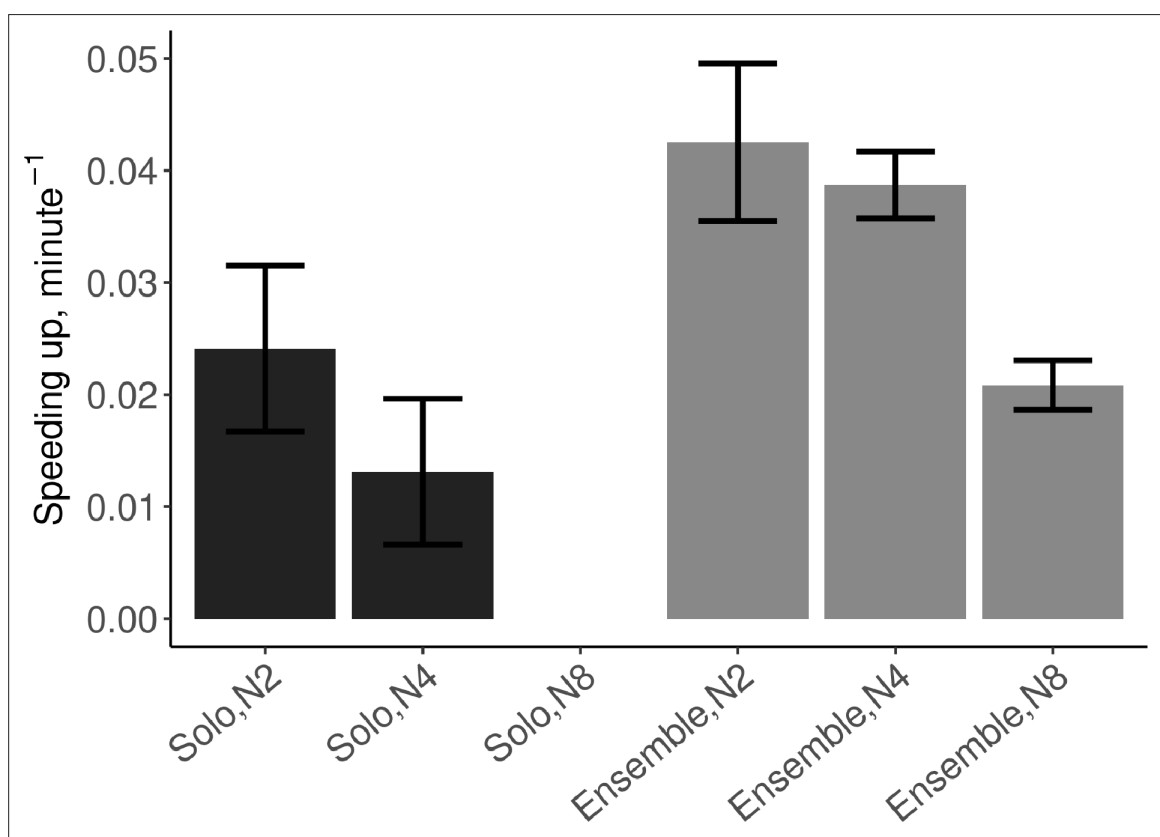

**Figure 4.** Mean (SE) speeding up defined as the linear slope of tempo over time, computed from the IOIs. N2=dyad; N4=quartet; N8=octet (no data collected in solo N8 condition). Error bars are standard errors. With missing trials, the number of observations in the respective conditions was n=(88, 111, Ø, 88, 110, 482).

The online version of this article includes the following source data and figure supplement(s) for figure 4:

**Source data 1.**

**Figure supplement 1.** Task performance in ensemble, synchronization-continuation task (SCT) trials, measured in terms of (**A**) speeding up, defined as the linear slope of tempo over time, and (**B**) variability.

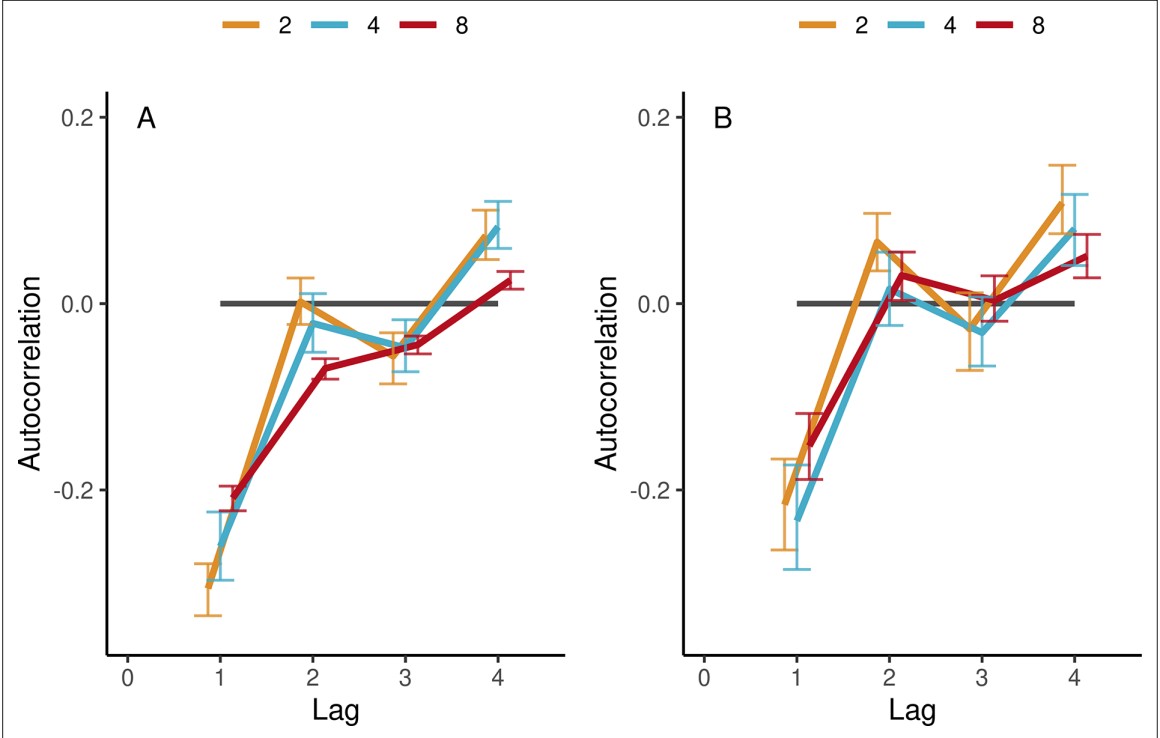

**Figure 5.** Autocorrelations in the synchronization-continuation task (SCT) in ensemble conditions. (**A**) Autocorrelations of individual IOIs, averaged (SE) across participants' trials and tempos, separately per group size (color-coded). (**B**) Same for group-aggregate IOIs. Error bars are bootstrap 95% confidence intervals. The number of observations in the respective group sizes was n=(88, 110, 474) in (A) and n=(44, 32, 66) in (B).

The online version of this article includes the following source data and figure supplement(s) for figure 5:

**Source data 1.**

**Figure supplement 1.** Autocorrelations of individual IOIs in ensemble synchronization-continuation task (SCT) drumming, averaged (SE) across individual participants' trials, shown separately per tempo (panels) and group size (color-coded lines).

was a trend across all group sizes for relative positive peaks at even lags 2, 4, 6, and 8 and relative negative peaks at odd numbered lags 1, 3, 5, and 7. The similarity of the dynamics of individuals and the group aggregate is not trivial because it implies that even groups of up to eight participants spontaneously acquire the alternating long-short interval dynamics characteristic of when individual participants synchronize with a stimulus.

### Inter-personal coordination (cross-correlations)

Cross-correlation applied to the pre-processed IOIs assessed inter-personal coordination between pairs of participants drumming together in the continuation phase of ensemble performances. On rare occasions, individuals produced diverging beat times by, for example, missing the drum or hitting it two times. The resulting outlier IOIs were removed, and the remaining were re-aligned relative to the other participants by temporal adjacency. The last pre-processing step consisted of whitening each time-series of IOIs (see From beat onset times to IOIs, outlier removal, and clustering–Pre-whitening).

We analyzed lags in the range 0–4 because cross-correlation coefficients tended to be symmetric between positive and negative lags, see **Figure 6**. As expected, cross-correlations at lag 0 were negative in duets ($\beta$=−0.039, SE=0.011, t=−3.60, and $\omega^2$=0.04), consistent with the pattern of results for autocorrelations, and inter-personal coordination dynamics were smoother in larger group sizes. Furthermore, lag 0 cross-correlations became less negative with increasing group size ($\beta$=0.012, SE=0.002, t=5.93, and $\omega^2$=0.13). Lag 1 cross-correlations were positive ($\beta$=0.14, SE=0.012, t=11.80, and $\omega^2$=0.32), but their magnitude decreased with group size ($\beta$=−0.016, SE=0.002, t=−7.14, and $\omega^2$=0.20). There were also effects of tempo, see **Figure 6—figure supplement 1**.

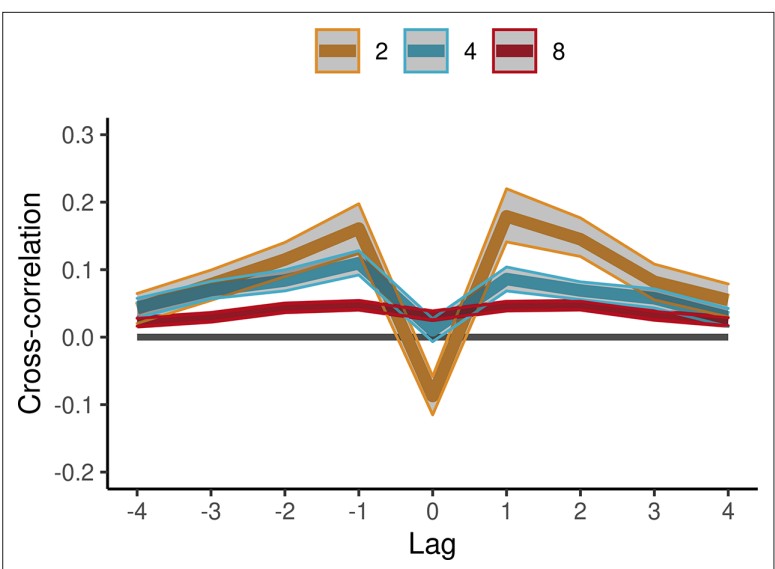

**Figure 6.** Cross-correlations in the synchronization-continuation task (SCT) in ensemble conditions. Averages (SE) across participant pairs and tempos are shown separately per group size (color-coded lines). IOIs were aligned across participants and pre-whitened by filtering with an autoregressive model. Error bars are bootstrap 95% confidence intervals. The number of observations in the respective group sizes was n=(44, 140, 1479).

The online version of this article includes the following source data and figure supplement(s) for figure 6:

**Source data 1.**

**Figure supplement 1.** Cross-correlations in the synchronization-continuation task (SCT) task, ensemble conditions, averaged across participant pairs (±CIs), per tempo (panels) and group size (color-coded lines).

## Group dynamics

### Network analysis

We used network analysis in the continuation phase of trials of ensemble playing to describe group coordination at an even higher level of organization than interpersonal coordination. First, we obtained the directed graphs of functional connectivity among participant drummers. In each trial, the graph consisted of the real-valued directed links between pairs of drummers estimated by way of the transfer entropy (TE), see *Figure 1C*. Then, network properties of these graphs were computed, see Network analysis. There was a significant effect of group size on causal density ($\beta$=−0.0006, SE=0.0001, t=−5.88, and $\omega^2$=0.47), see *Figure 7A*. Causal density increased with tempo ($\beta$=0.008, SE=0.0019, t=4.01, and $\omega^2$=0.12). As *Figure 7B* shows, mean node strength increased with group size ($\beta$=0.0032, SE=0.00032, t=10.13, and $\omega^2$=0.73). Node strength was affected by tempo ($\beta$=0.016, SE=0.005, t=2.96, and $\omega^2$=0.07). For better model conditioning, tempos values were reduced and compressed by transforming them with a logistic function. Comparing between SCT trials and synchronization-only trials in the octet group found no difference for causal density (t<1) or mean node strength (t<1), suggesting group dynamics were similar regardless of whether a pacing stimulus was present or not.

### Network properties and task performance

The relation between group dynamics and group performance was evaluated by regressing the network properties separately with respect to speeding up and IOI variability in ensemble playing. Speeding up decreased with increasing causal density ($\beta$=−0.005, SE=0.002, t=−2.22, and $\omega^2$=0.02) and was not associated with node strength ($\beta$=−0.01, SE=0.007, and t=−1.66), see *Figure 7—figure supplement 1A–B*. Variability was not associated with causal density ($\beta$=−.003, SE=0.0017, and t=−1.97) but decreased with increasing node strength ($\beta$=−0.007, SE=0.0019, t=−4.01, and $\omega^2$=0.09), see *Figure 7—figure supplement 1C–D*. Interestingly, we also observed that TE tended to be higher in trials that deviated less from the stimulus tempo, see *Figure 7—figure supplement 2*.

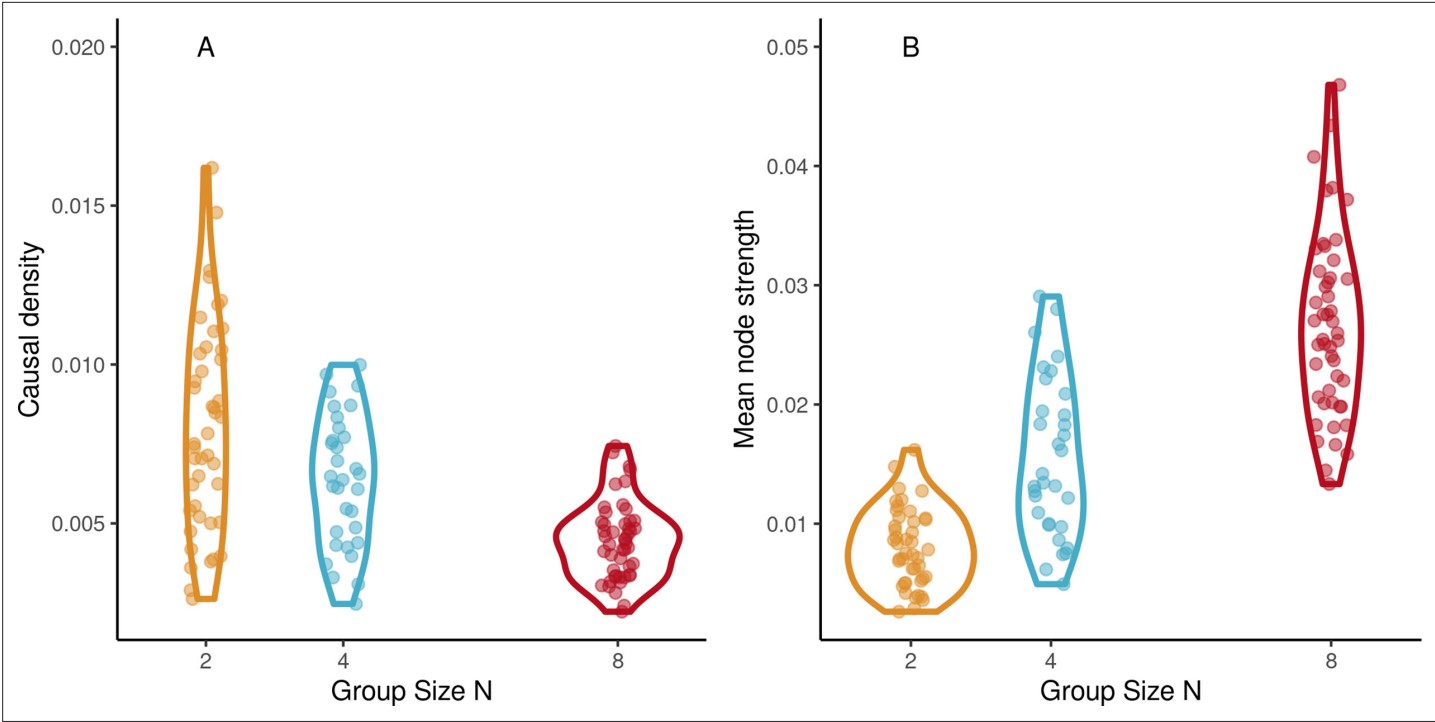

**Figure 7.** Network dynamics. (**A**) Mean node strength increases with group size in the continuation phase of synchronization-continuation task (SCT) ensemble drumming trials. (**B**) Causal density decreases with group size (Abscissa jittered for visibility).

The online version of this article includes the following source data and figure supplement(s) for figure 7:

**Source data 1.**

**Figure supplement 1.** Association between drumming performance measures and network properties in synchronization-continuation task (SCT) ensemble trials.

**Figure supplement 2.** Transfer entropy (TE) is highest in synchronization-continuation task (SCT) trials with minimal tempo increase relative to the instructed tempo.

## Variability in the mean-field model with discretely updated feedback

The results of the modeling showed that the cycle duration variabilities of the mean-field hybrid Kuramoto model, adapted to have discretely updated feedback, reproduced the main experimental findings. The model results are summarized in *Figure 3B* and can be compared to the experimental results in *Figure 3A*. The adapted model reproduced the pattern of behavioral results at both the level of individual oscillators (relating to individuals in our drumming data) and the mean field (relating to our group-aggregate analyses). Specifically, for all group sizes, variability was higher for individual oscillators when playing in the group than when playing solo, but variability was always lowest for the mean field (group aggregate). In addition, with increasing group size, the difference between individual variability and that of the mean field became greater, indicating increasing collective benefits with increasing group size. See *Figure 3—figure supplement 1B* for a comparison with the null (pseudo-ensemble) condition. A separate report will address the model's individual unit dynamics and co-variation in more detail (Delasanta et al., in preparation). The importance of the mean-field hybrid model for understanding the present results can be seen in that other versions of the Kuramoto model did not successfully replicate the pattern of the data. Specifically, a pulse-coupled network in a neighbors-only ring-topology and a constant coupling network in full connectivity did not exhibit the same pattern of variability as the empirical data, see *Figure 3—figure supplement 2A–B*.

## Discussion

The present study examined how group performance in a synchronization timing task depends on group size and interactions among group members. Our overall hypothesis was that the mean field

would influence individuals in a group, thereby stabilizing group performance. We expected that larger groups would exhibit more stable performance due to a more stable mean field. This hypothesis was motivated by the collective dynamics that allow animals with limited capacity for interacting with each other to achieve mutual goals by exploiting processes such as positive collective feedback and wisdom of crowds (*Sumpter, 2006*). As expected, we observed that temporal variability in group drumming decreased with group size. This occurred only at the level of the ensemble or mean field when the group was taken as a single entity. The effect was reversed for individual participants; their variability increased from solo to group playing conditions and with group size. We showed that this result agrees with a collective dynamics framework based on an adapted Kuramoto model. The model is consistent with the idea from theoretical biology that self-organizing systems can successfully coordinate with an external constraint by reducing their disorder at the macro-level at the expense of increasing their micro-level disorder (*Kugler and Turvey, 1987*; *Prokopenko et al., 2014*) and that medium-sized groups exhibit commonalities with principles of organization found both in two-unit and large collectives (*Kelso, 2021*; *Zhang et al., 2019*). Our theory is related to the Vicsek model, an influential account of collective animal behavior which explains the spontaneous coherent heading direction arising in collective animal movement, also in human crowds, in terms of local interactions of attraction and repulsion (*Silverberg et al., 2013*; *Vicsek et al., 1995*). The Vicsek model also relies on individual coupling with the average of a group, with the important difference that only a small radius of neighbors defines the interaction field, called a local order parameter (*Chaté et al., 2008*).

The continuous-discrete Kuramoto system introduced here included fluctuating intrinsic frequencies and event-based feedback. Consistent with the discrete nature of time asynchronies in a drumming task, the model used a pulse function instead of continuous coupling. The implications of this idea reach beyond the topic of the present study. The combination of continuous movement and discrete sensory sampling of the environment is a frequently observed scenario; in the present study, participants used the discrete auditory feedback of drum taps to inform their continuous movements leading up to future drum taps. Our task reduced the opportunity of visual coupling but did not eliminate it completely, leaving the possibility that group drumming involved both discrete and continuous feedback such as visual information about continuous arm, hand, and drumstick movements related to the timing of individual drumming sounds. Future studies should address whether visual feedback during group drumming is informative, as well as explore how that visual information is captured within the individual and collective dynamics of the system.

Interestingly, in the octet group, individual variability was similar in both the synchronization-only condition, where the metronome pacing stimulus was present, and in the self-paced continuation stage of the SCT trials, where no pacing stimulus was present. This result implies that, although tempo accuracy and drift were superior when the pacing stimulus was present, the stability of group timing in a group of eight participants may hit a ceiling, where performance becomes as stable as with a pacing stimulus.

Individuals' tempos tended to increase in solo trials, and as expected, this effect was amplified (i.e. speeding up was greater) when trying to stay synchronized with other participants (*Okano et al., 2017*). Surprisingly, the amount of speeding up was not higher for larger groups of drummers, and it was even reduced for larger groups, at least at faster tempos. In contrast, accounts based on mutual prediction imply more speeding up for larger groups (*Thomson et al., 2018*; *Wolf et al., 2019*). This discrepancy highlights the importance of testing tempo drift with specific task constraints and avoiding hasty generalizations from smaller to larger groups. Indeed, tempo can also slowdown in some circumstances. This was observed, for example, in a group synchronization task when participants synchronized their oscillations using swung pendulums, a task involving continuous visual coupling rather than discrete acoustic event coupling (*Bardy et al., 2020*).

Having summarized task performance in terms of consistency and accuracy, we next considered drumming dynamics. Individuals' autocorrelations exhibited negative coefficient peaks at odd lags and positive coefficient peaks at even lags. This is the typical error-correction process demonstrated when individuals synchronize their tapping with a stimulus or with each other in pairs (*Konvalinka et al., 2010*; *Repp, 2005*). With increasing group size, however, the magnitudes of the autocorrelation coefficient fluctuations with lag decreased; individuals' alternating patterns became smoother. This suggests that as group size increases, drumming dynamics are dominated less by tap-by-tap error-correction and more by a group dynamic. Interestingly, the group-aggregate onsets exhibited the

same autocorrelation pattern as individuals, even in octets. This implies that the collective mean field behavior spontaneously acquired some of the dynamic properties typically associated with individual performance.

Inter-personal coordination in duets exhibited the negative cross-correlation at lag 0 and positive cross-correlation at lag 1 expected from past studies (*Repp, 2005*). The results for speeding up, variability, and cross-correlations in the dyads were consistent with the literature on dyadic interaction with tapping tasks. Yet, as group size increased in the present study, the coefficient magnitudes for cross-correlations decreased. In fact, they all but disappeared for groups of eight, suggestive of a leader-less group coordination scenario (*Heggli et al., 2019*). In a small number of trials, we even observed positive lag 0 cross-correlations in groups of eight, a signature of anticipation emerging from dynamic interaction. This result is consistent with the idea that group members coordinated less with each other and more with the mean field. Along with a more slowly decaying autocorrelation function, this suggests the emergence of a larger-scale group dynamic in the octet. As we hypothesized, mutual prediction is not necessarily the only basis for coordination in larger groups. The present model would be more like a predictive account had we included a new variable and an additional equation in the model, $\dot{\theta}_{j,i} = f\left(\theta_i, \theta_j, \theta_{j,i}\right)$, corresponding to a separate internal oscillatory process $\theta_{j,i}$ that individual $i$ dedicates to the prediction of an external signal $j$. Yet, there is no reason to assume that collective dynamics and mutual prediction are exclusive alternatives, only that collective dynamics are a mode of social interaction that emerges under certain constraints such as large groups of interacting agents.

Characterizing inter-personal interaction on the basis of bivariate measures is typical in the context of dyadic paradigms. Larger groups, however, create the possibility for higher-order interactions. Network analysis was applied to the connectivity graph of TEs among drummers. Larger groups had lower causal density but higher node strength. This is possible because individual nodes in larger groups project to a larger number of other individual nodes and suggests that in larger groups, each individual has a decreased influence on the group, both less potential to destabilize the group and lower requirements to keep the group stable. We also found that higher node strength was associated with better overall performance, both in terms of variability and tempo accuracy.

Group timing tasks address some of the foundations of human social interaction. This area of research has been labeled the 'dark matter' of social neuroscience (*Dumas et al., 2014*; *Insel, 2010*; *Schilbach et al., 2013*) because it has not received enough attention despite the fact that large parts of the human cortex respond to social stimuli (*Frith and Frith, 2001*). The temporal regularities in music make it an ideal stimulus to promote synchronous and cooperative movement, providing evidence as to why music is often present at emotional social gatherings such as weddings, funerals, parties, and political rallies. Feelings of affiliation and cooperation increase between people who have experienced synchronous movement with each other, even during infancy (*Cirelli et al., 2014*; *Cirelli et al., 2016*; *Hove and Risen, 2009*; *Mogan et al., 2017*; *Trainor and Cirelli, 2015*; *Valdesolo et al., 2010*). The ability of music to strengthen social bonds within a group may have evolutionary roots that enhance survival capabilities of the group (*Huron, 2001*; *Savage et al., 2020*; *Trainor, 2015*). Arguably, the presence of a convergent regular rhythm defined over the ensemble allows individuals of varying skill level to perform together without having to predict each note they play, an idea that remained largely untested in the present study because of the relative simplicity of the isochronous stimulus.

Participants in group activities sometimes report a so-called state of flow characterized by absorption, loss of sense of control, loss of self-consciousness, togetherness, and effortless action toward a shared goal (*Csikszentmihalyi, 1990*; *Gaggioli et al., 2017*; *Hart et al., 2014*). Little is known about the underlying group processes that enable such experiences. Some of the required conditions have been studied in the context of group musical performance (*Bishop et al., 2019*; *D'Amario et al., 2018*). Collective intentionality is often seen as an extrapolation of single-person cognitive processes in that individuals predict other individuals' goals and actions (*Tomasello, 2000*, *Tomasello, 2014*). Yet, the neural and cognitive mechanisms historically identified in single-person paradigms do not always map onto the ontology of social behavior (*Przyrembel et al., 2012*). Collective intentionality can stand for an independent mode of social cognition (*Satne and Salice, 2018*; *Zahavi and Satne, 2015*). Social cognition theory can even be turned upside-down by observing cases where individual agency arises from social interaction (*De Jaegher and Froese, 2009*; *Froese et al., 2014*). The present work shows how collective dynamics can absorb individual action.

## Beyond the group average

We have argued that attraction to the group average constitutes an organizing principle that supports the coordinated performance of medium and large groups of individuals. There are theoretical and empirical reasons, however, to assume that other organizing tendencies can compete with the group average and potentially lead to more complicated dynamics. To begin with, our synchronization task was designed to foster collaboration, did not challenge individual skill, and did not create any incentives for competitive individual behavior. This collaborative aspect corresponds to social behaviors where crowds want to appear as a strong unity, for instance, when chanting a protest song or storming into battle. In contrast, choir singing demonstrates that both cooperation and competition are involved often in group musical performance (*Keller et al., 2017*). And different players in musical ensembles often simultaneously play different parts with different melodic and rhythmic features. While these parts must fit together, reflecting cooperation, at any one point, the player of the 'melody' or most important part influences other players more than vice versa, setting up leadership dynamics within the group (e.g. *Chang et al., 2017*). Different cultural contexts and performance styles may even emphasize the creative and expressive de-synchronization from the regular group beat, or avoid isochrony altogether, allowing musicians to unfold their own ideas or belonging to a specific tradition (*Benadon et al., 2018*; *Davies et al., 2013*; *Lucas et al., 2011*). Understanding musical creativity requires a situated approach and ethnographic engagement to find which features of the emergent collective acoustic environment constitute meaningful affordances for given musicians in a given context (*Linson and Clarke, 2017*).

The combination of attractive and repulsive coupling among individuals is particularly evident in the context of collective animal behavior. The well-known Vicsek model accounts for the coherent heading direction in collective movement in terms of a combination of local attraction and repulsion forces (*Vicsek et al., 1995*). In chorusing animals, there is a tendency for male individuals to cluster their acoustic behavior in time but also to advance ahead of the cluster, presumably to have better chances at attracting the attention of other relevant individuals (*Gamba et al., 2016*; *Greenfield et al., 2017*; *Ravignani et al., 2014*; *Ravignani et al., 2019*).

The present study focused on a scenario with uniform coupling by keeping group members close to each other and restricting the timbre and pitch of their drums to a very narrow range, making them acoustically inseparable from each other when playing synchronously. Musicians who play on large stages sometimes report noticeable delays amongst each other. In the present study, the delay was limited in the range from 4 ms to 16 ms depending on partners' spatial separation, and the sound intensity drop was from 6.3 to 15.1 dB. Yet, selective attention plays an important role in animal chorusing (*Greenfield et al., 2021*). Variations in the layout of the acoustic environment, individualized timbres, and the availability of alternative coupling modalities such as vision, all create affordances for selective attention and individualized roles in the ensemble. For example, low-pitch sounds afford better beat-based timing precision in humans (*Burger et al., 2013*; *Hove et al., 2014*; *Hove et al., 2020*). The discernability of individual sounds facilitates temporal coordination tasks, presumably through the involvement of neural mechanisms for self-other segregation and integration mechanisms (*Liebermann-Jordanidis et al., 2021*; *Novembre et al., 2016*; *Ragert et al., 2014*).

In conclusion, we assume that the aforementioned factors fit within the collective dynamics framework as symmetry-breaking terms, complications in task space that enable more interesting stabilities than mere attraction to the group average. Future studies can address this idea by manipulating individual heterogeneity, task difficulty, spatial layout, and coupling topology.

## Materials and methods

### Participants

A total of 102 university students participated in the duet (n=11 sessions; 22 participants), quartet (n=8 sessions; 32 participants), and octet (n=6 sessions; 48 participants) groups. In addition, 88 high-school students participated in the octet condition (n=11 sessions). Participants were recruited from the department's participant pool with course credit and from local high schools as part of science field trips to the university. Across the set of all participants, musical training was distributed in the range from non-musician to amateur musician. We did not control the composition of musical skill within each ensemble.

## Apparatus

In ensemble playing conditions, participants were seated in chairs facing each other, equally spaced in a linear, square, or octagon arrangement, with diagonal distances of 3 meters in the duet and quartet groups and 6 meters in octet group. Ensemble playing conditions were performed in a large, tall-ceiling dance studio, part of the LIVELab, a laboratory dedicated to auditory and movement neuro-science (http://livelab.mcmaster.ca). In the duet and quartet groups, solo trials were recorded with individual participants immediately before or after the ensemble sessions in an adjacent sound-proof room.

In the duet and quartet groups, participants used drumsticks on plastic buckets that were damp-ened and placed upside-down. Their fundamental frequencies varied in a narrow range (117–129 Hz). They are of the type seen in street-style group drumming observed in the community and were used previously by a group drumming workshop instructor. Piezoelectric sensor elements were taped on the inner side of the attack surface of each bucket. The analog signal was recorded on a computer at 48 kHz via a multi-channel studio-grade analog-digital interface (Focusrite RedNet 4) and the digital audio production software Reaper (Cockos Incorporated, Rosendale, NY, USA). This setup provided for a clean signal with minimal noise and a sharp, high-amplitude attack that was digitized into onset times in Matlab using a custom algorithm with thresholding and rejection of rebounds under 150 ms.

The octet group used a set of electronic pads which, unlike the bucket drums, allowed tuning the drums to distinct fundamental frequencies across a larger number of individual drums. Eight iden-tical pads (Yamaha TP70 7.5" snare/tom) and eight independent powered studio reference monitors (Yorkville Sound YSM5) placed under each pad were connected to the head module (Yamaha DTX 920 K). The pads had linear amplitude response and, on impact, produced the same woodblock sound. The woodblock fundamental frequencies, about double those of the bucket drums, also varied in a narrow range (249–284 Hz). The two setups were similar as they both involved polymer impact surfaces and piezoelectric sensors embedded within those surfaces to detect impacts and record onset times. To record the electronic drums, the head module was connected to the computer which stored drum hits as MIDI events. The recording line was parallel to the drum set and did not contribute additional latency between drum impact and drum sound (5 ms). Separately, we measured that the recording pipeline added negligible timing variability with a SD of 1 ms relative to the ground truth based on a direct analog recording of the drum pad impacts.

The reference metronome that participants were asked to synchronize with was played over a dedi-cated speaker placed in the middle between participants. A custom patch for the sound synthesis and production software Pure Data (*Puckette, 1997*) ran on the computer to generate and time a sharp electronic kick drum sound, with a timbre very distinct from the individuals' drums (600 Hz sine wave with 250 ms duration, 20 ms amplitude attack, and 230 ms decay). The stimulus was fed back and recorded as an additional channel along with the participants' drum beats.

## Stimuli

The stimulus tempo of the isochronous reference metronome varied between 50 and 240 beats per minute (bpm) depending on condition (see below). Trials were 90 s long, and the synchronization phase always included 32 stimulus events. Thus, slower tempos had longer duration synchronization phases.

## Task

Drumming is ecologically valid in terms of common musical practice, and tapping with sticks has been shown to improve timing relative to finger tapping (*Madison et al., 2013*; *Manning et al., 2017*). The task on each trial was a SCT. During the synchronization phase, participants were presented with trials containing a reference isochronous metronome at a particular tempo and were asked, using drumsticks, to: (1) synchronize with the given reference metronome, (2) keep their own variability as low as possible, (3) keep their drumming rate as steady as possible, and (4) synchronize with the other drummers when in an ensemble condition. During the continuation phase, the reference metronome stopped, and participants were told to continue to keep their drumming rate as steady as possible while synchronizing with other drummers in an ensemble. Those in the octet group also completed trials in a synchronization-only control task in which the isochronous metronome played for the entire trial (i.e. there was no continuation phase). Participants were instructed to focus their eyes on an 'X'

taped in the centers of their drums, thus reducing the visual coupling among group members in the ensemble conditions.

## Design

Participants were assigned to one of three group sizes: duet (2), quartet (4), or octet (8). Those in the duet and quartet groups completed the SCT task in both solo and ensemble conditions. Duet and quartet groups completed six trials at each of the following tempos: 50, 80, 120, 160, 200, and 240 bpm (or 1200, 750, 500, 375, 300, and 250 ms onset-to-onset intervals between beats) in each of the solo and ensemble conditions, with the order of tempos randomized. For each set of participants, there was random assignment to either the solo or ensemble condition first. Trials lasted 90 s with a synchronization phase of 32 metronome beats.

For octet groups, there were three differences in the design. First, they did not complete solo trials because of time and space constraints. Second, the set of tempos was slightly different (80, 100, 120, 160, and 200 bpm or 750, 600, 500, 375, and 300 ms intervals). Finally, an additional condition was included, consisting of a synchronization-only control task (i.e. with the reference metronome throughout).

We found that the very low and very high tempos led to poor performance; hence, we eliminated extreme tempo trials and only analyzed the set of matching tempos across dyads, quartets, and octets, namely 80, 120, 160, and 200 bpm (750, 500, 375, and 300 ms). See the figure supplements for the extreme tempo conditions. For all groups, prior to completing the trials in each condition, participants engaged in practicing synchronized steady drumming without an initial metronome to set the tempo so that participants could become familiar with the sticks and drums and playing together in an ensemble.

## Procedure

After providing informed consent, filling out demographic and music background questionnaires, reading instructions, and observing a demonstration of the apparatus, participants were instructed to find a comfortable posture on their chair and adjust the drum position during practice. They were asked not to adjust these further during the trials. Participants were allowed to practice as needed to feel comfortable with the task. After each trial, the experimenter verified with the participants that they were ready to start drumming again before the next trial was initiated. The whole session from arrival to departure lasted about an hour. Before asking for informed consent, the overall goals of the study were explained to the participants, including that the collected drumming data would potentially be presented anonymously at meetings and in scientific papers, and that the procedures had been approved by the McMaster Research Ethics Board.

During the solo performance in the duet condition, one participant was moved to an adjacent sound-proof room while the second remained in the large dance studio. For the quartet condition, two of the participants were randomly assigned to arrive earlier and perform their solo trials before the group trials and the other two to perform their solo trials after the group trials. In the octets, individuals did not perform solo trials.

## Analysis

### Pre-processing of onset times

Onset times were processed using custom Matlab scripts (an example script for cleaning and matching IOIs is available at https://gitlab.com/dodo_bird/group_sync_elife, copy archived at swh:1:rev:6479ed2e3409ebc156fbc917a246cf4c4edf4f44, *Dobromir, 2022*). In SCT trials, only data from the continuation phase was analyzed. In the synchronization-only control trials of the octet group, the full-trial length was analyzed. The following steps were taken to remove artifacts and condition the data according to the requirements of the analyses used.

### Signal

On a few occasions, participants accidentally pulled the cable from the drum and disconnected or damaged the sensor. A total of 15 bad channels (i.e. participants) were detected and removed from further analysis with a custom script using the onset variability, signal noise, and amplitude consistency. For further verification, we visually inspected the signals and onset times from all channels and trials.

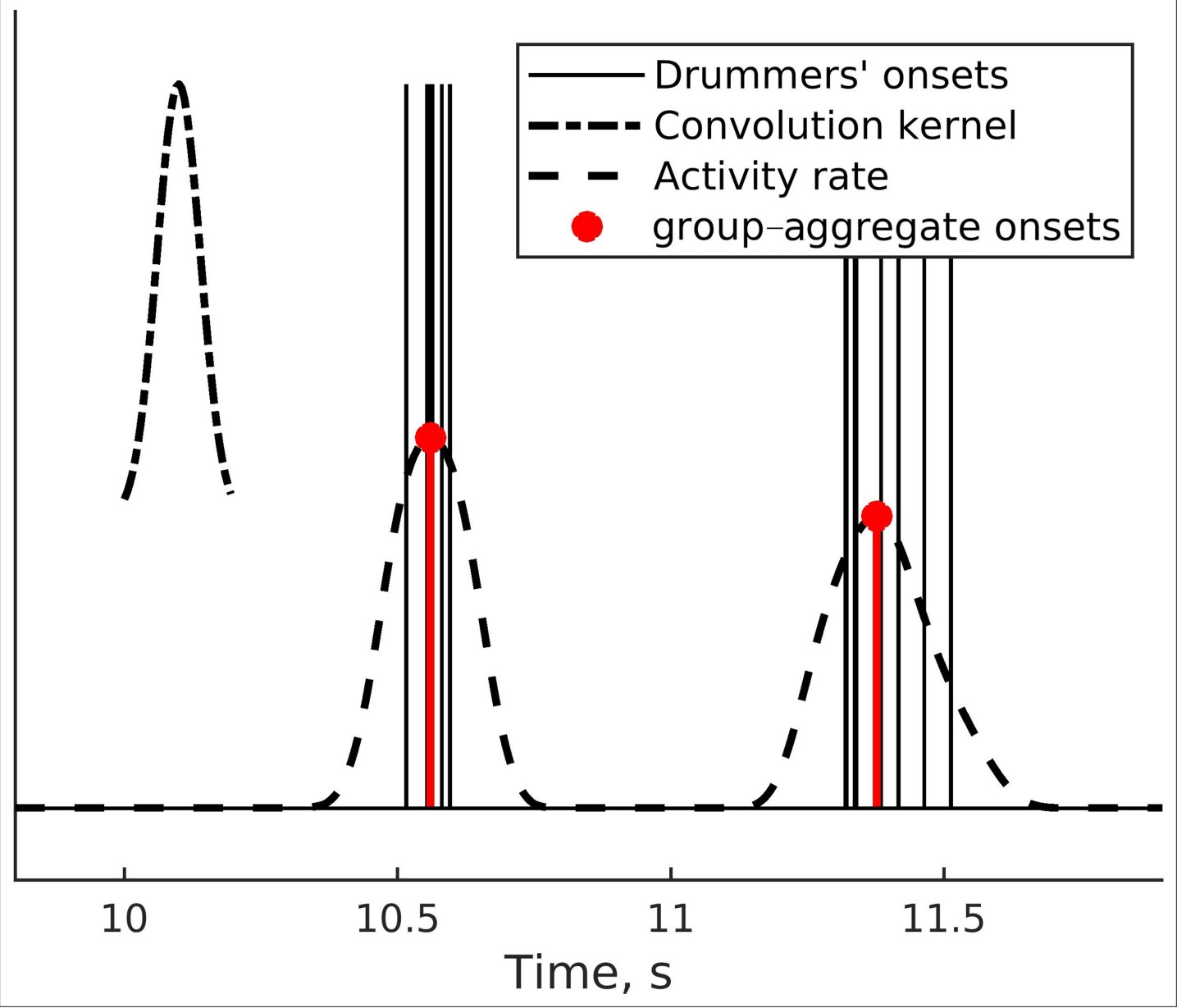

**Figure 8.** Group-aggregate onset times from convolving the drummers' onsets with a Gaussian kernel.

### Group-aggregate (mean field) measure for ensembles and pseudo-ensembles

For ensemble trials, we created a measure of the beat onset times of the group as a whole by constructing theoretical group-aggregate onsets in the center of clusters of individuals' onset times, $t_{n,g}$, where $n$ is the beat onset number, and $g$ signifies it is the group-aggregate onset. Specifically, we: (1) convolved onset times with a Gaussian kernel 200 ms in width, (2) smoothed the resulting group activity rate with a moving average window of a quarter of the reference stimulus beat onset-to-onset time, and (3) took the peak locations for group-aggregate beat onset times (see **Figure 8**). This algorithm, typically used in neuroscience to extract a firing rate or population activity from the spike times of multiple recorded units (**Dayan and Abbott, 2001**), provides a measure of the central tendency of one or more events clustered in time. The method does not necessarily restrict the group-aggregate pattern to one average onset because if a single onset time is isolated away from other clusters, for example, when a participant erroneously hits the drum while the other participants were silent, it

will also be detected as a single-event cluster. Such events of large individual deviations from the group were very rare in the present study and eliminated from analysis because, overall, participants performed the task consistently and in agreement with group members. Group-aggregate beat onset times were calculated for each trial in each group during ensemble playing conditions. In solo trials of dyads and quartets, participants could not see or hear each other, but we computed onset times of the pseudo-group-aggregate for a baseline.

### From beat onset times to IOIs, outlier removal, and clustering

For each trial for each participant, the sequence of onset times $t_{n,i}$, where $i$ is participant and $n$ onset number, was differenced to produce a time-series of inter-onset-intervals, $IOI_{n,i} = tn_{n,i} - t_{n-1,i}$. Where appropriate for a more intuitive interpretation, we expressed these as tempos, $T_{n,i} = 60/IOI_{n,i}$. The same procedure was carried out to obtain the time-series of IOIs for the group-aggregate measure in ensemble conditions and the pseudo-aggregate measure in solo playing conditions.

If a participant skipped a beat on a certain occasion, this would produce an outlier IOI twice as large as most others in the trial, thus biasing subsequent measures. IOIs exceeding 50% difference above or below the trial median were removed. The measures of individual variability and tempo trend (see below) were based on these IOIs.

### Alignment of IOIs

Quantifying inter-personal coordinated tapping with the cross-correlation requires pairing IOIs from all participants in a given trial. To this end, we used a custom algorithm to detect IOIs that were matching across all group members. To illustrate, if one group member skipped a beat, the produced IOI was about double that of the others, and the corresponding IOIs for all group members were removed.

### Pre-whitening

Cross-correlations are difficult to interpret in the presence of non-stationarities such as slow drift up and down of the IOI. We applied a method for removing such non-stationarities by pre-whitening the IOIs (*Dean and Dunsmuir, 2016*). Specifically, each time-series of IOIs was filtered using an auto-regressive model with coefficients estimated with automatic model-order selection ('pre-whiten' function from the time series analysis package TSA for R).

## Performance measures and dynamics

All subsequent measures were based on the individual participants' timing data. For the sake of characterizing overall group performance, the group-aggregate timing was analyzed in the same way as was individual participant timing and reported as a separate condition.

### Variability and tempo trend

The tempo trend was defined as the linear trend of IOIs along successive beats in a trial. The slope $b$ in the regression equation $T_{n,i} = a + b_{ni}$ was fitted separately in each trial for each individual. The variabilities of each individual and group-aggregate were estimated separately using the coefficient of variation of their de-trended and pre-whitened IOIs.

### Autocorrelation

Individuals' and group-aggregate dynamics were analyzed using autocorrelation of IOIs.

## Inter-personal coordination

### Cross-correlation

The temporal pattern of co-variation among participants was analyzed using cross-correlation applied to the pre-whitened IOIs.

### Transfer entropy

The bidirectional interaction between every pair of participants in a group was measured using TE (*Schreiber, 2000*). This is an information-theoretic measure of directed mutual information used for quantifying the effective connectivity in, for example, moving animals (*Brown et al., 2020*), sensori-motor and motor-motor interactions (*Dotov and Froese, 2018*; *Lungarella and Sporns, 2006*;

*Stoffregen et al., 2009*), and complex networks of neurons (*Gourévitch and Eggermont, 2007*). Effective connectivity, sometimes also referred to as causality, is understood in the Wiener sense that having information about past source dynamics improves the prediction of future target dynamics (*Wiener, 1956*). We used a Matlab toolbox for TE (*Ito et al., 2011*) adapted for point processes such as neuronal firing times where a delay $d$ separated the source ($J$) and target ($I$) dynamics are binary events, $TE_{J \to I}(d) = \sum p(i_{t+1}, i_t, j_{t+1-d}) \log_2 \frac{p(i_{t+1}|i_t, j_{t+1-d})}{p(i_{t+1}|i_t)}$. The $t$ indices here are bins of time, set to 10 ms. We took the maximum TE over a range of up to 100 delay steps (i.e. 1 s). We performed the analysis either with or without removing the trend of decreasing intervals between time indices in a trial (i.e. trend for tempo to speed up) and found this did not change the pattern of results.

Arguably, TE has several benefits over comparable methods (*Lungarella et al., 2005*). We identified another reason for applying TE to the sequence of time onsets $t$ rather than what would be the first choice historically, namely TE or Granger causality among sequences of IOIs paired across participants. The latter scenario involves several steps of subtractive pre-processing consisting of removing outliers due to missed taps, pruning and aligning observations across participants, and filtering smooth trends that could be intrinsic to the performance, whereas applying a version of TE for point processes (sequences of onsets) avoids the steps which risk excessive data removal.

## Network analysis

The measures in Performance measures and dynamics are univariate and describe drumming performance of individuals. This individual-level analysis does not say much about how participants coordinate with each other to achieve group performance. Inter-personal coordination lists bi-variate measures typically used to analyze dyadic tapping and sensorimotor coordination problems. Yet, even such pair-level analysis is not sufficient in the group context where more than two participants coordinate simultaneously with each other. Hence, we obtained group-level measures by analyzing the network of effective connectivity among drummers (*Newman et al., 2011*). Drummers were treated as nodes (or vertices) and TE values as the weights of links (or edges) of a directed graph.

## Causal density and mean node strength

Among the rich set of phenomena that potentially can be quantified in a large network, only a few measures are relevant in the present context because of the low number of nodes and the real-valued nature of the graph consisting of continuous connection weights rather than binary links. We defined causal density as the average of all edge weights (*Seth et al., 2011*; *Seth and Edelman, 2004*). Mean node strength (*Barrat et al., 2004*), obtained using the igraph package for R (*Csardi and Nepusz, 2006*), was defined as the average across drummers of each drummer's total outbound connectivity. This is a version of the more popular measure of node degree which is used for binary graphs.

## Statistical analyses

Linear mixed effects models (*Bates et al., 2015*; *Singer and Willett, 2003*) were used instead of ANOVAs because of the complex, unbalanced design and missing observations. By taking different subsets of the data and fitting separate models, we tested for: (1) effects of group size in SCT trials comparing dyads, quartets, and octets, (2) differences between solo and ensemble playing conditions in dyads and quartets, (3) differences between SCT and synchronization-only in octets, and (4) differences between individual and group-aggregate performance. For brevity, only the selected model with best fit is reported in Results. The modeling procedure was incremental, starting with the minimal specification of a grand mean and iteratively including predictors and their interactions until the model became over-specified or too complex to converge, which tended to be the case when including tempo. The Satterthwaite method was used to determine significant effects and interactions (*Kuznetsova et al., 2017*). Effect sizes are given by the $\omega^2$ (*Ben et al., 2020*).

## Model parameters

We simulated 100 runs per condition and each run corresponded to a trial with duration of 100 s. We used a sampling rate of 300 Hz and a third-order solver. Different sampling rates and using solvers with the Euler method or with an adaptive step size produced similar results. Performance variability was measured in terms of the coefficient of variation of cycle durations, treating zero-phase crossings as onset times, reproducing the behavioral data qualitatively. The pulse in *Equation 5* with parameters

$a$=1.25 and $b$=0.02 specifies an asymmetric gamma distribution that rises sharply at time 0 and then decays slowly in about 200 ms. The code needed to run the full simulation is available at https://gitlab.com/dodo_bird/group_sync_elife.

Conditions consisted of the full crossing of group size $N$=(2,4,8) with coupling strength ($K$=0 in solo and $K$=8 in ensemble). The noise term had a mean of zero and variability σ=6. Intrinsic frequencies $\omega_i$ were drawn from a normal distribution with a mean of 2 Hz and SD of 0.5. Comfortable finger tapping in humans is in the range 1.5–2 Hz (*McAuley et al., 2006*), tapping is optimally precise in the range 0.8–2.5 Hz (*Moelants et al., 2002*), and walking cadence on average is at 2 steps/s (*MacDougall and Moore, 2005*).

## Acknowledgements

We would like to thank Daniel Bosnyak, Sally Stafford, Susan Marsh-Rollo, and LIVELab staff. This research was supported by grants to LJT from the Social Science and Humanities Research Council of Canada, the Natural Sciences and Engineering Research Council of Canada, the Canadian Institutes of Health Research, and the Canadian Institute for Advanced Research.

## Additional information

### Funding

| Funder | Grant reference number | Author |
| --- | --- | --- |
| Social Sciences and Humanities Research Council of Canada | Insight Grant 435-2020-0442 | Laurel Trainor |
| Natural Sciences and Engineering Research Council of Canada | RGPIN-2019-05416 | Laurel Trainor |
| CIFAR | | Laurel Trainor |

The funders had no role in study design, data collection and interpretation, or the decision to submit the work for publication.

### Author contributions

Dobromir Dotov, Conceptualization, Data curation, Software, Formal analysis, Methodology, Writing – original draft, Writing – review and editing; Lana Delasanta, Formal analysis, Writing – original draft, Writing – review and editing; Daniel J Cameron, Conceptualization, Methodology, Writing – review and editing; Edward W Large, Conceptualization, Software, Formal analysis, Supervision, Writing – original draft, Writing – review and editing; Laurel Trainor, Conceptualization, Formal analysis, Supervision, Funding acquisition, Methodology, Writing – original draft, Writing – review and editing

### Author ORCIDs

Dobromir Dotov http://orcid.org/0000-0002-5543-360X
Daniel J Cameron http://orcid.org/0000-0001-5543-9836
Edward W Large http://orcid.org/0000-0003-3909-3518
Laurel Trainor http://orcid.org/0000-0003-3397-2079

### Ethics

Before asking for informed consent, the overall goals of the study were explained to the participants, including that the collected drumming data would potentially be presented anonymously at meetings and in scientific papers, and that the procedure had been approved by the McMaster Research Ethics Board, protocol #2164.

### Decision letter and Author response

Decision letter https://doi.org/10.7554/eLife.74816.sa1
Author response https://doi.org/10.7554/eLife.74816.sa2

## Additional files

### Supplementary files

• Supplementary file 1. Supplementary table. Separate linear mixed-effects models were fitted for each of lags 1–8 of the autocorrelations of individuals (a) and group aggregates (b) in the continuation phases of synchronization-continuation task (SCT) trials, ensemble conditions. The predictors were group size ($N$) and *Tempo* (80, 120, 160, and 200 beats per minute [bpm]). For simplicity, the same full model was fitted in all cases for individuals, $Y_{ik}=(\beta_0+\sigma_{0g})+\beta_1 N_{ik}+\beta_2 Tempo_{ik}+\beta_3 N_{ik}Tempo_{ik}+\sigma_g$, where $i$ is participant, $k$ trial, and $g$ group. For the group aggregate (b) where there was not enough data to fit models with all predictors, $Y_{ik}=(\beta_0+\sigma_{0g})+\beta_1 N_{ik}+\beta_2 Tempo_{ik}+\sigma_g$ was used. Significant coefficients (p<.05, Satterthwaite method) are in bold.

• Transparent reporting form

### Data availability

The current manuscript is a computational study, so no data have been generated for this manuscript.

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
