## [Editor Report]

Taking joint drumming as a model of collective dynamics, and combining solid quantitative methods, the authors characterize how human behavior changes, at the individual- and group-level, as a function of group numerosity. A take-home message of this important work is that not everything we know from studies involving dyads should be necessarily generalized to larger groups. This study will be of great interest to scientists looking for new approaches to understanding group behavior, especially within the fields of human cognition, neurosciences, and musicology.

---

## [Decision Letter]

**Decision letter after peer review:**

Thank you for submitting your article "Collective dynamics support group drumming, reduce variability, and stabilize tempo drift" for consideration by *eLife*. Your article has been reviewed by 3 peer reviewers, and the evaluation has been overseen by a Reviewing Editor and Aleksandra Walczak as the Senior Editor. The following individual involved in review of your submission has agreed to reveal their identity: Michael Greenfield (Reviewer #2).

Essential revisions:

1) The data do fit, to some extent, the proposed hypothesis (response to the mean of all neighbors). However, as noted by Reviewer #2, there are alternative hypotheses which may be as good or better.

2) The fact that the 'agents' could see as well as hear each other poses some problems for adapting the Kuramoto model correctly. Moreover, selective attention (to only some neighbors) has not been dealt with in the proposed hypotheses or in the perceptual and response mechanisms of the individual agents. The issue that individuals might have adapted more strongly to the output of their neighbours should be discussed in the article.

3) Both reviewers #1 and #3 recommended that the article should clarify which of the conclusions from the study might be generalisable to musical activities.

4) Statements regarding predictive mechanisms should be fully justified. Note also the point made by Reviewer #1: the task used here requires minimal prediction due to the constant tempo of the task.

*Reviewer #1 (Recommendations for the authors):*

Here are some comments and suggestions.

1. Abstract. I would try to be more explicit about potential dissociations between dyadic vs. large group interactions (i.e. comparing this study with previous ones focusing on dyads). The authors might even attempt to refer to this issue more explicitly in the abstract.

2. Introduction, page 4, "arising from pressures to either collaborate or compete". The study by Keller et al., 2017 (Frontiers in Psychology), showing that collaboration and competition are not always alternatives in joint music making, might also be relevant here.

3. Introduction, page 4, "Group music making constitutes an ecologically valid and convenient paradigm for studying group action and collective experience in the laboratory". This statement would be better supported by a reference, e.g. D'Ausilio et al., 2015 (TICS).

4. Introduction. The role of "prediction" in "interpersonal synchronization" is discussed briefly on two occasions in the introduction (page 4 and page 5, final sentence). I am not sure the authors have sufficiently conveyed their message. I believe they should better explain how prediction would emerge "implicitly" from the dynamic models they are referring to. Ideally, they should provide a concrete example of this, and support this with some data from previous studies. Otherwise, I am not sure a naïve reader will be able to make much of this issue.

5. Methods. The equipment used across small and large groups is slightly different. I don't think this should be a problem. However, I believe it would be good to explain why this was the case. This might be important for colleagues planning other studies in the future.

6. Methods and results. Page 9: "We found that the very low and very high tempos led to poor performance, hence we eliminated extreme tempo trials and only analyzed the set of matching tempos across dyads, quartets, and octets, namely 80, 120, 160, and 200 bpm". I believe it would be important to show this data (ideally as a supplement, i.e. not in the main text).

7. Methods and results. Section 2.7.2.1. "In ensemble conditions, we only kept the group-aggregate tempo trend". For comparability with previous studies testing smaller groups, it might be important to conduct this analysis looking at each individual separately, and then average the results. I realize this might yield qualitatively similar results, but I still think it's an important detail to report somewhere.

8. Methods and results. Page 19. "IOIs were matched across participants by proximity". Does this mean that cross correlations were computed between pairs of participants that were physically proximal? Or have the authors computed these coefficients by cross-correlating each individual's IOIs with the aggregate IOI measure? This is an important detail, which I am not sure I have fully understood.

9. General comment. The authors have selectively tested experimental conditions in which the effects produced by the performers were qualitatively similar (i.e. same acoustic output). I would like the authors to make a prediction about what would happen should the effects produced be different. Considering that the authors often refer to music, and that musical ensembles are often composed of players producing distinct sounds, I believe it might be important to discuss this.

10. General comment. Following on the previous point, the authors might consider that in this experiment participants only needed to "integrate" the acoustic outputs produced by the other players. However, in more complex coordination dynamics (e.g. an orchestra), each individual might need to weight the outputs produced by the other members, eventually "integrating" with some and "segregating" from others. This principle is well known in auditory cognition and has also been proposed to be important for ensemble music performance (see e.g. Ragert et al., 2014 (Plos One), Novembre et al., 2016 (Neuropsychologia), Liebermann-Jordanidis et al., 2021 (Acta Psychologica)). Could the proposed model account for this kind of behavior? And, if so, how? Would this require modelling individuals as separate processes (see also point 4 above)?

11. General comment. In relation to the role of "prediction", the authors state that this "is implicit in the emergent synchronization of coupled individual oscillatory dynamics". I am not sure this can be generalized to all instances of interpersonal coordination though. Take, as an example, the instance where one of the drummers gradually changes tempo (I am not referring to microtiming deviations or phase adjustments, but to an actual gradual change of the period). This might strongly call for predictive processes, more so than in the current scenario (where there is not much to predict given that all participants have to drum at a stable, non-changing tempo). A more ecological example would be joint musical improvisation, where players notably introduce unexpected behaviors, yet their peers are able to synchronize with (and sometimes even anticipate) these. To what extent can the discussed results and model be generalized to similar instances?

*Reviewer #2 (Recommendations for the authors):*

Several details not addressed above:

It is noted that the human subjects filled out a questionnaire which included information on musical experience/background. Was this information used to eliminate or retain subjects? Would musical background, or lack thereof, potentially influence the results of drummer timing? Some detailed information on this issue would be helpful.

The paragraph at the bottom of page 4, beginning with 'There are models of coordination where prediction... ' could benefit from a better explanation. There are some profound statements here, and it would behove the authors to avoid possibilities of misinterpretation.

*Reviewer #3 (Recommendations for the authors):*

Please, have a look at the "groove" and expressivity studies (groove has many culturally specific definitions: swing, swingue, ginga, balanço, etc) and reflect on how these studies solve and shed light on some aspects of the contribution. In special, understand that cultural forms profit from intentional and idiomatic variability and not simply avoid them. In my opinion, the ontological stance of the musical phenomenon as a creative act violates important assumptions behind traditional statistical methods, and researchers must take care to avoid building Procrustean beds of predictability in a context where predictability is not welcome.

Discuss and explain how procedures could have affected the shape and magnitude of timing variability.

Explain the axes, indicate magnitudes and units, and define ALL elements, terminologies, and abbreviations presented in formulas and figures. Repeat explanations in the captions. The proper visualization of time in the graphs is important because the stimuli designed with 20 ms attack slope and perception times add 10ms or more in the thresholds of perception. Therefore the clear indication of IOIs or another timing reference in terms of milliseconds helps to understand the results.

Explain the procedures in the attempt to clarify the chain that could add delays (latency) in the system: sampling, midi protocols, soundboards, Since we are dealing with timing issues and small perception thresholds, small delays and their nature (constant, variable, random) greatly affect the validity of measurements.

Verify if your statistical/numerical methods are not violated by homogeneity and independence (e.g. a delay in tempo is often compensated by the performer in the following attacks).

Try to communicate and describe the musical/performative effect of the results to open a dialogue with researchers in the music field.

---

## [Author Response]

Essential revisions:1) The data do fit, to some extent, the proposed hypothesis (response to the mean of all neighbors). However, as noted by Reviewer #2, there are alternative hypotheses which may be as good or better.

We agree and we included a discussion of alternative mechanisms and scenarios (bottom of page 3, the bottom of page 8, the end of first paragraph on page 16, the end of paragraph 1 on page 18, end of paragraph 2 on page 15, and of course the whole new section 3.1). We consider the mean field approach to be the most likely for the present scenario because the task makes it hard to pay selective attention to individual participants. We discussed the role of individual heterogeneity and spatial separation in breaking symmetry and creating affordances for more interesting forms of coordination. An additional argument is provided by comparing different versions of the model. We included a neighbour-only coupling (ring topology) version of the model which does not correspond well to the empirical data. We showed this in Figure 3—figure supplement 2B.

2) The fact that the 'agents' could see as well as hear each other poses some problems for adapting the Kuramoto model correctly. Moreover, selective attention (to only some neighbors) has not been dealt with in the proposed hypotheses or in the perceptual and response mechanisms of the individual agents. The issue that individuals might have adapted more strongly to the output of their neighbours should be discussed in the article.

We included further discussion of this limitation of the task. Indeed, we cannot rule out visual coupling which is of a different nature than the short auditory pulses. We also included the following points. First, we emphasized that participants were instructed to focus on X’s in the center of their drums, hence partners were only visible in the peripheral field (pages 22-23, section 4.4). Second, our results in the dyad group essentially replicated other dyadic tapping studies (end of page 17). Third, we included two new versions of our model, one with a continuous coupling (classical Kuramoto), and one with a restricted topology, a ring, where individual units are only coupled to their immediate neighbours (bottom of page 8 and Figure 3—figure supplementary 2). The classical model would be expected to fit the data if the primary input was continuous (i.e., visual) and the ring model would be expected to fit the data if participants were indeed focusing on the visual information from those beside them. However, neither model fit the empirical data well.

3) Both reviewers #1 and #3 recommended that the article should clarify which of the conclusions from the study might be generalisable to musical activities.

We agree that the present task is an idealized impoverished scenario. We inserted a more thorough discussion of the limitations of the present study and where future research can be directed to capture more realistic musical activities in a new section, “Beyond the group average”.

4) Statements regarding predictive mechanisms should be fully justified. Note also the point made by Reviewer #1: the task used here requires minimal prediction due to the constant tempo of the task.

We agree that the isochronous task is not challenging as a pattern. Because we did not design the study to specifically test predictive processes, we prefer to minimize the discussion of predictive mechanisms. It is hard to talk about predictive processes and whatever is the alternative to such processes without engaging in a lengthy discussion including definitions and how prediction can be used to mean different things. Having said that, this has been an important topic of research in the last few years that should not be ignored. To this end, we made revisions so that the topic is acknowledged but without committing to uncovering new knowledge about prediction as our study was not designed to do that. We do now, however, pointed out a definite way for distinguishing models in that context. Specifically, one set of models, typically associated with predictive processing theories, assume that two separate (oscillatory) processes are needed to account for an individual: one for controlling one’s own timing, and the second for tracking an external process (the partner). Our approach makes use of only one oscillator per individual; doesn’t separate the mechanism into self-timing and other-tracking. We added a brief mention of this distinction at the beginning of page 18.

Reviewer #1 (Recommendations for the authors):Here are some comments and suggestions.1. Abstract. I would try to be more explicit about potential dissociations between dyadic vs. large group interactions (i.e. comparing this study with previous ones focusing on dyads). The authors might even attempt to refer to this issue more explicitly in the abstract.

Thank you for the suggestion, we have now done this.

2. Introduction, page 4, "arising from pressures to either collaborate or compete". The study by Keller et al., 2017 (Frontiers in Psychology), showing that collaboration and competition are not always alternatives in joint music making, might also be relevant here.

Thank you for the suggestion. We included this at a different location (bottom paragraph of page 19), in the final section where we discuss other potential principles of organization that come into play in tasks with both competitive and cooperative aspects.

3. Introduction, page 4, "Group music making constitutes an ecologically valid and convenient paradigm for studying group action and collective experience in the laboratory". This statement would be better supported by a reference, e.g. D'Ausilio et al., 2015 (TICS).

Thank you for the useful suggestion. We had added this (beginning of paragraph 2, page 5).

4. Introduction. The role of "prediction" in "interpersonal synchronization" is discussed briefly on two occasions in the introduction (page 4 and page 5, final sentence). I am not sure the authors have sufficiently conveyed their message. I believe they should better explain how prediction would emerge "implicitly" from the dynamic models they are referring to. Ideally, they should provide a concrete example of this, and support this with some data from previous studies. Otherwise, I am not sure a naïve reader will be able to make much of this issue.

Indeed, our introduction of this issue was very cursory in an attempt to avoid turning this manuscript into a comparison of predictive and non-predictive theoretical models of coordination. We made changes so as to acknowledge the existence of predictive accounts without committing to them as our study was not designed to specifically examine predictive processes (last paragraph of page 4 and paragraph 1 on page 18).

5. Methods. The equipment used across small and large groups is slightly different. I don't think this should be a problem. However, I believe it would be good to explain why this was the case. This might be important for colleagues planning other studies in the future.

The main reason was circumstance. Additionally, while the drums in the initial setup had distinct sounds, they didn’t allow us to manipulate them to produce eight drums with nominally different fundamental frequencies. Thus, when the eight-drum set became available we switched to it.

6. Methods and results. Page 9: "We found that the very low and very high tempos led to poor performance, hence we eliminated extreme tempo trials and only analyzed the set of matching tempos across dyads, quartets, and octets, namely 80, 120, 160, and 200 bpm". I believe it would be important to show this data (ideally as a supplement, i.e. not in the main text).

We edited the relevant figures and captions in the figure supplements and inserted the data from all tempos.

7. Methods and results. Section 2.7.2.1. "In ensemble conditions, we only kept the group-aggregate tempo trend". For comparability with previous studies testing smaller groups, it might be important to conduct this analysis looking at each individual separately, and then average the results. I realize this might yield qualitatively similar results, but I still think it's an important detail to report somewhere.

We verified that this does not make a difference, see our response above. For simplicity, we replaced this with the stats based on individual data.

8. Methods and results. Page 19. "IOIs were matched across participants by proximity". Does this mean that cross correlations were computed between pairs of participants that were physically proximal? Or have the authors computed these coefficients by cross-correlating each individual's IOIs with the aggregate IOI measure? This is an important detail, which I am not sure I have fully understood.

We edited to be clearer (first paragraph of 2.2.2, page 12). What was meant here was not spatial proximity with the neighbours but deciding which drum hits from different participants go together and which ones, on very rare occasions, are mistakes. The issue of spatial proximity, raised by the other reviewers as well, is very important for understanding large ensembles but is less relevant here because of the relative proximity of all drummers (see also section 3.1. “Beyond …”).

9. General comment. The authors have selectively tested experimental conditions in which the effects produced by the performers were qualitatively similar (i.e. same acoustic output). I would like the authors to make a prediction about what would happen should the effects produced be different. Considering that the authors often refer to music, and that musical ensembles are often composed of players producing distinct sounds, I believe it might be important to discuss this.

We believe that introducing acoustic heterogeneity, as well as difficulty, will create affordances for selective attention. For example, low-pitch sounds may afford better beat-based timing precision. In general, heterogeneity should make it possible for specialized individual roles, sub-groups and leader-follower dynamics to emerge. See 3.1. “Beyond the group average” at the end of the Discussion.

10. General comment. Following on the previous point, the authors might consider that in this experiment participants only needed to "integrate" the acoustic outputs produced by the other players. However, in more complex coordination dynamics (e.g. an orchestra), each individual might need to weight the outputs produced by the other members, eventually "integrating" with some and "segregating" from others. This principle is well known in auditory cognition and has also been proposed to be important for ensemble music performance (see e.g. Ragert et al., 2014 (Plos One), Novembre et al., 2016 (Neuropsychologia), Liebermann-Jordanidis et al., 2021 (Acta Psychologica)). Could the proposed model account for this kind of behavior? And, if so, how? Would this require modelling individuals as separate processes (see also point 4 above)?

Thank you for the relevant literature and ideas. We have incorporated the literature, bottom of page 20. Additionally, we now discuss that the present approach needs to be expanded to be able to account for competitive pressure and selective attention. See also our discussion of the well-known Vicsek model (middle of page 16 and second paragraph of page 20).

11. General comment. In relation to the role of "prediction", the authors state that this "is implicit in the emergent synchronization of coupled individual oscillatory dynamics". I am not sure this can be generalized to all instances of interpersonal coordination though. Take, as an example, the instance where one of the drummers gradually changes tempo (I am not referring to microtiming deviations or phase adjustments, but to an actual gradual change of the period). This might strongly call for predictive processes, more so than in the current scenario (where there is not much to predict given that all participants have to drum at a stable, non-changing tempo). A more ecological example would be joint musical improvisation, where players notably introduce unexpected behaviors, yet their peers are able to synchronize with (and sometimes even anticipate) these. To what extent can the discussed results and model be generalized to similar instances?

We agree. We removed the idea of implicit prediction and discussed other, richer forms of performance that are not really tested by the present paradigm. Please see our responses to similar comment above and below from the other reviewers.

Reviewer #2 (Recommendations for the authors):Several details not addressed above:It is noted that the human subjects filled out a questionnaire which included information on musical experience/background. Was this information used to eliminate or retain subjects? Would musical background, or lack thereof, potentially influence the results of drummer timing? Some detailed information on this issue would be helpful.

We have inserted a short description of this aspect of the demographics, stating that the sample consisted of participants who all had some musical and dance experience without extreme outliers. We believe that expertise will interact with the task especially if it is more difficult, creating the opportunity for more interesting dynamics including clear asymmetries between participants.

The paragraph at the bottom of page 4, beginning with 'There are models of coordination where prediction... ' could benefit from a better explanation. There are some profound statements here, and it would behove the authors to avoid possibilities of misinterpretation.

We edited this part as suggested here, and also by Reviewer 1.

Reviewer #3 (Recommendations for the authors):Please, have a look at the "groove" and expressivity studies (groove has many culturally specific definitions: swing, swingue, ginga, balanço, etc) and reflect on how these studies solve and shed light on some aspects of the contribution. In special, understand that cultural forms profit from intentional and idiomatic variability and not simply avoid them. In my opinion, the ontological stance of the musical phenomenon as a creative act violates important assumptions behind traditional statistical methods, and researchers must take care to avoid building Procrustean beds of predictability in a context where predictability is not welcome.

We agree. A lot about musical expression is left out from the present study. We now discuss the limitations related to issues of context, individual heterogeneity, and the ability to resist the group average, and encourage future research to incorporate the influence of the mean field as one factor that is likely necessary for understand the range of musical behaviours.

Discuss and explain how procedures could have affected the shape and magnitude of timing variability.Explain the axes, indicate magnitudes and units, and define ALL elements, terminologies, and abbreviations presented in formulas and figures. Repeat explanations in the captions. The proper visualization of time in the graphs is important because the stimuli designed with 20 ms attack slope and perception times add 10ms or more in the thresholds of perception. Therefore the clear indication of IOIs or another timing reference in terms of milliseconds helps to understand the results.

We tried to make corresponding changes but we found there is very little in the figures and results that is actually defined at the millisecond scale. Since we are not focused on asynchronies but variability of the intervals, the CV is the only really relevant measure that reflects sub-second variations but it is a dimensionless variable.

Explain the procedures in the attempt to clarify the chain that could add delays (latency) in the system: sampling, midi protocols, soundboards, Since we are dealing with timing issues and small perception thresholds, small delays and their nature (constant, variable, random) greatly affect the validity of measurements.

We included more detail about this to clarify that the recording was parallel to sound triggering, hence participants did not experience added delays between surface contact and sound onset, other than the minimal latency of the professional electronic drum set which is about 5 ms.

With respect to timing variability resulting from the chain from the electronic drums module and USB connection to MIDI-recording on the computer, we tested the apparatus specifically for the sake of the present revisions and created Author response image 1. It summarizes the timing error from the ground truth based on a direct analogue recording of the piezoelectric elements. It shows that the digital pipeline adds very little timing variability. On very few occasions it drops drum hits, and how we handled these was already explained in the methods.

**Author response image 1. sa2fig1:** 

Verify if your statistical/numerical methods are not violated by homogeneity and independence (e.g. a delay in tempo is often compensated by the performer in the following attacks).

Thank you for the thorough suggestions. The main assumption of the statistical method (linear mixed effects models with maximum likelihood estimation) is normality of the residuals and random effects of the given fitted model. We included examples of how we verified that this assumption is satisfied to a reasonable degree. For example, Author response image 2 show how the given data fit a normal distribution. The first row is for the residuals of the model for tempo speeding up. Second row, the random effects from the same model (the random effects estimate individuals’ or groups’ overall tendency to speed up aside from effects of experimental condition). Left, model fitted to individuals. Right, model fitted to the group-average of individuals for better statistical properties.

Try to communicate and describe the musical/performative effect of the results to open a dialogue with researchers in the music field.

Thank you for encouraging us to do so. We hope the responses to the previous comments are sufficient to also address this important one.[Disp-formula equ4 equ5][Disp-formula equ4 equ5]